

# The effect of post-activation enhancement on the performance of Chinese national skeleton athletes in the "ice push sled"-a first cross-sectional study

Guang Tian*, Haojie Li*, Huan Zhu and Binghong Gao

School of Athletic Performance, Shanghai University of Sport, Shanghai, China
* These authors contributed equally to this work.

## ABSTRACT

**Objective:** To investigate the impact of post-activation potentiation (PAP) induced by resisted sled sprint at different loads on the subsequent 30 m ice push sled performance of Chinese skeleton athletes, and to identify the resisted sled sprint load that most effectively enhances PAP for Chinese skeleton athletes.

**Methods:** Seven elite athletes from the Chinese skeleton team participated in four tests with more than 48 h intervals. During the tests, on the first test, athletes completed a 40 min standard warm-up, rested for 6 min, and then performed a 30 m test. On the second, third, and fourth test, athletes completed the standard warm-up, then performed 20 m sprints with resisted sled (RS) at 75%, 50%, and 25% of body mass (BM), respectively, rested for 6 min, and then performed the 30 m test.

**Results:** No significant differences were found in morning pulse, blood urea, and creatine kinase levels among four tests. The percentage of maximum heart rate (%HRmax) within different intensity ranges showed no significant differences among four tests. However, significant differences were observed in ice push sled performance among four tests (No BMRS: 5.08 ± 0.27; 25% BMRS: 5.05 ± 0.29; 50% BMRS: 5.02 ± 0.27; 75% BMRS: 5.04 ± 0.28). *Post hoc* analyses revealed that the 50% BMRS test had faster speed compared to the no resistance ($p < 0.05$), the 25% BMRS ($p < 0.05$), and the 75% BMRS ($p < 0.05$) tests. Additionally, the 75% BMRS test had faster speed than the no resistance test ($p < 0.05$).

**Conclusion:** A 20 m sprint with 50% BMRS effectively enhances the PAP effect in skeleton athletes, improving their ice push sled performance. Coaches can incorporate this resisted sled sprint in athletes' training routines for performance enhancement in both daily training and pre-competition preparations.

# INTRODUCTION

Skeleton is a winter sport that originated in the late 19th century and is now one of the official sports of the Winter Olympic Games, after years of development (*Chun & Park, 2020*; *Millet, Brocherie & Burtscher, 2021*). It is characterized by high speed and high risk; participating athletes are required to be in prime physical test and have incredible technical

Corresponding author
Binghong Gao,
19901730313@163.com

skill (*Steffen et al., 2017*). Skeleton has become increasingly popular in China, attracting a greater number of young people to participate. The intensity of the sport has attracted much attention in the Winter Olympics. Its high-speed racing rhythm and exciting course design attract many spectators and have become an important part of the Winter Olympic Games. Skeleton is one of the important programs of China's winter sports team, and has made significant contributions to China's development in the field of ice and snow sports (*Hu, Wang & Liu, 2022*).

The performance of skeleton competition are determined by both the ice push sled performance and the sled sliding performance on the ice track (*Bullock et al., 2009b*). The push sled phase marks the beginning of the race, with the key factor being how athletes utilize push sled techniques to influence their initial speed and starting performance (*Needham et al., 2021*). According to *Zanoletti et al. (2006)*, the technical proficiency in push sled significantly impacts the competition results. High-quality push sled techniques not only generate higher speeds at the start but also provide better stability during the curve sections, which is crucial for the progress and outcome of the race. This phenomenon highlights the push sled capability as a significant factor influencing competitive performance in skeleton. A fast and effective push sled start is a prerequisite for achieving success in the race, significantly enhancing the athlete's performance and final results throughout the track (*Colyer et al., 2018b*).

There is a clear relationship between sprinting ability and push sled performance in skeleton (*Colyer et al., 2018b*). Sprinting ability refers to a capacity of athlete to accelerate rapidly over a short period, which is directly related to the starting speed in the push sled phase. Research indicates that elite push sled athletes typically exhibit higher performance in sprint and vertical jump tests, which can serve as important predictors of their push sled performance (*Sands et al., 2005*). For example, in Australia's skeleton selection program, an assessment of four successful female athletes showed significantly faster 30 m sprint times in the initial screening phase (*Bullock et al., 2009a*). This data suggests that they possess superior capabilities in the push sled phase. Such ability not only facilitates a strong start during the push sled phase but also provides substantial support for the subsequent sliding phase, further validating the decisive role of sprinting ability in push sled performance.

The post-activation potentiation (PAP) strategy has been used to improve acute sprint performance (*Zisi et al., 2022*; *Stavridis et al., 2023*). Studies by *Winwood et al. (2016)* and *Cross et al. (2017)* showed that heavier loads-75% of body mass-resisted sled training-led to subsequent improvements in sprinting performance. In skeleton, one related study has shown that using a 50% of body mass sled to resist a 20 m full sprint effectively improves the PAP of skeleton athletes, which improves their 30 m sprint performance (*Tian et al., 2022*). However, the ice push sled in skeleton is quite different from land sprinting (*Zhu, Gong & Tian, 2022*) and it is important to train to improve an athlete's performance in ice push sled. Since the short-distance sprinting ability is the most important athletic quality of skeleton pushers in the initial phase, training with the PAP strategy may be helpful to

Tian et al. (2024), *PeerJ*, DOI 10.7717/peerj.18271 2/14

improve the ice push sled performance of skeleton athletes. However, due to the influence of weather tests (snow, light), ice temperature, blade temperature, start slope and start sequence (*Han, Shi & Wu, 2021*), there is a big difference between ice push sled and land sprinting. In addition, although the initial position of the skeleton start is similar to that of the sprint start, skeleton athletes do not lift their torso during the push sled, which is inconsistent with the sprint, and the angle of the torso does not change; skeleton athletes are required to demonstrate the ability to generate force at high speeds when jumping onto the sled (*Colyer et al., 2018a*), and athletes are required to push a sled weighing 30–40 kg very quickly (*Colyer et al., 2018b*). This requires a combination of strength and speed. Ice push sled start performance has a more direct effect on skeletal performance than the performance of athlete at a 30 m sprint, and has a greater impact on overall race performance.

At present, there is no research on the effect of PAP on the starting ice push sled performance after resisted sled sprinting. It is important to understand the impact of PAP with different loads of sled resistance to improve the performance of ice push sled, which has important practical significance for the Chinese athletic team. Therefore, Chinese national skeleton athletes who were preparing for the Winter Olympic Games were selected for this study. This article investigates the impact of PAP strategies induced by resisted sled sprint at different loads on the subsequent 30 m ice push sled performance of Chinese skeleton athletes, and to identify the resisted sled sprint load that most effectively enhances PAP. This study is the first of its kind and has great practical significance for the development of the skeleton program, specifically for the Chinese athletes.

## PARTICIPANTS AND METHODS

### Participants

The seven elite snowmobile athletes of the Chinese national team constitute a special population for this study. Despite the small sample size, these athletes have a high degree of consistency and specialization in their competitive level. The main goal of the study was to investigate the effects of resisted sled sprints with different loads on the athletes' subsequent ice-pushed sled performance. Despite the limited sample size, our statistical analyses revealed significant differences, suggesting that training under different loading tests has a significant effect on athlete performance.

A total of seven Chinese national skeleton athletes were selected as research subjects, four of whom participated in the Beijing Olympic Winter Games. All seven athletes made up the core team members of the Chinese skeleton program and are contenders for the Winter Olympics. All subjects had no lower limb or back injuries in the 6 months prior to the study. All athletes completed the test at the same time of day, taking into account their daily biorhythms. During the testing period, all athletes wore the same racing clothes and shoes. All athletes signed an informed consent form, which was approved by the Ethics Committee of Shanghai Sport University (No.102772020RT081). The basic information of the athletes is shown in Table 1.

**Table 1 Basic information of athletes.**

| Participant | Gender | Height/cm | Weight/kg | BMI/kg·m$^{-2}$ | Age/year | Campaign level |
|---|---|---|---|---|---|---|
| 1 | Male | 183.7 | 82.3 | 24.4 | 26 | IC |
| 2 | Male | 178.9 | 76.6 | 23.6 | 27 | IC |
| 3 | Male | 177.2 | 84.6 | 26.9 | 27 | IC |
| 4 | Female | 171.9 | 63.9 | 21.6 | 21 | IC |
| 5 | Female | 169.6 | 61.9 | 21.5 | 25 | IC |
| 6 | Female | 172.8 | 65.9 | 22.1 | 25 | IC |
| 7 | Female | 168.3 | 68.9 | 24.3 | 23 | IC |

**Note:**
IC, International champion.

## Methods

### PAP induction test procedure

In this study, athletes were required to complete four tests, each with an interval of 48 h or more. The warm-up and PAP induction protocols were conducted on the plastic track in the warm-up room, and the ice push sled test was conducted on the ice course in the igloo at the National Sliding Centre. Athletes were assessed for fitness prior to the start of each test. Each of the four tests began with a 40 min standard warm-up. The warm-up included a 10 min jog to increase cardiovascular endurance and overall body temperature. Dynamic stretching followed (Stretching time 5 min), targeting major muscle groups including the hamstrings, quadriceps, calves, and hip flexors. This was followed by specific dynamic sprinting exercises consisting of 20 m sprints at intensities of 90%, 95%, and 100% with 2 min rest intervals between sprints (*Whelan, O'Regan & Harrison, 2014*).

For the first test, athletes proceeded to the ice push sled test after a 6 min rest following the warm-up. For subsequent tests, the standard warm-up was completed, and then 75%, 50%, and 25% of the body mass was used to perform a 20 m resisted sled sprint, respectively, and then the ice push sled test was performed after a 6 min rest (Fig. 1). In this study, the order in which the resistance loads were tested was not randomized. The order of testing was 75% of body mass for the second test, 50% of body mass for the third, and finally 25% of body mass. In the resisted sled sprint, the weight of the sled was calculated based on each athlete's body mass and was adjusted accordingly before each test. The adjustment of the load on the sled was achieved by means of additional weight plates to ensure that the total weight of the sled, including the sled itself and the added counterweight, met the experimental requirements. These weight plates were securely fastened to the sled with straps and clips to ensure stability during the resisted sled sprint.

### Ice push sled test

Athletes wore race suits (Anta, Jinjiang, China), gloves (Anta, Jinjiang, China), ice spikes (Anta, Jinjiang, China) and helmets (BMW, Munich, Germany). Athletes started in the individual competition push-skate starting position when the start signal for the push-skate course was lit and remained consistent throughout the four tests. During the test, the track was maintained, cleaned and iced by the professional ice maker of the Beijing

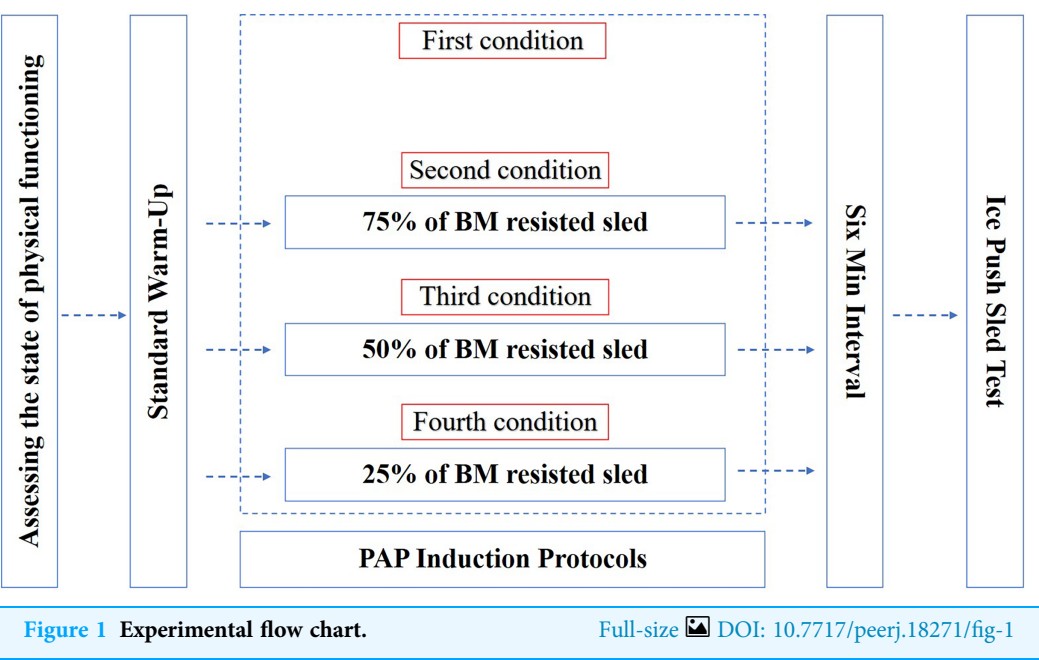

**Figure 1 Experimental flow chart.**

Winter Olympics Yanqing track to ensure that the tests remained basically the same during the test. Athletes were verbally encouraged throughout the push sled to ensure maximum effort for each repetition. Timing was done with the OMEGA T3000 timing system (OMEGA, Zurich, Switzerland). The ice push sled test site and timing system were consistent with those used in the Beijing Winter Olympics. The environment was similar across all four tests, with an average temperature of −4.4 °C, an average humidity of 32%, and an average ice temperature of −8.1 °C.

### Blood urea, creatine kinase test

To avoid potential interference with the experimental results due to changes in the athlete's physiological test, blood urea (BU) and creatine kinase (CK) were measured using a Roche Dry Biochemistry Analyzer (Reflation Plus, Japan). A total of 80 μL of fingertip blood was collected from the athlete's fingertip at 6:30 am on the morning of the test day, after the athlete had rested sufficiently on the day before the test.

### Morning pulse and heart rate test during warm-up

Heart rate (HR) testing included a morning pulse test and a warm-up HR test, to avoid potential interference with the experimental results due to variations in the athlete's physiological test and warm-up intensity. The morning pulse test was performed with a finger clip oximeter (YX306; Yuwell, Fujian, China) with the athlete fully rested the day before the test, and the test was performed at 6:30 am on each test day. A warm-up HR test was performed with a Polar heart rate monitor (Team Pro; Polar, Kempele, Finland) during the 40 min warm-up. Then, the real-time warm-up HR was divided by the maximum heart rate obtained in advance from the trainer to get the percentage of maximum heart rate (%HRmax).

## Statistical analysis

Data were analyzed using SPSS24.0 statistical software (Chicago, IL, United States) for statistical processing, and results were expressed as mean ± standard deviation (M ± SD). The Shapiro-Wilk method was used to test whether the data followed a normal distribution. One-way repeated measures ANOVA was used to analyze the ice sled pushing performance, morning pulse, morning BU and CK, and warm-up HR data as a whole, and *post hoc* comparisons were performed using the least significant difference (LSD) method for two-way comparisons, and $\eta^2$ was used to calculate the effect size (ES), with a significance level of $p < 0.05$. ES quantifies the proportion of variance in the dependent variable that is explained by the independent variable, with $\eta^2$ values around 0.01 indicating a small effect, 0.06 a medium effect, and 0.14 a large effect.

# RESULTS

## Results of morning pulse, morning start BU and CK before resistance testing of sled with different loads

As shown in Table 2, one-way repeated measures ANOVA on the four pre start test morning pulse, morning rise BU and CK values showed no significant difference between the morning pulse (F = 0.254, $p = 0.857$, ES = 0.207), morning rise BU (F = 0.838, $p = 0.490$, ES = 0.375) and CK (F = 0.602, $p = 0.622$, ES = 0.316) values.

## Results of heart rate monitoring during the warm-up phase of sled resistance with different loads

As can be seen in Table 3, a one-way repeated measures ANOVA analysis of the warm-up phase heart rate values during the four tests showed that %HRmax was ≤60% (F = 1.331, $p = 0.295$, ES = 0.472), the %HRmax was 60–70% (F = 0.529, $p = 0.668$, ES = 0.297), the %HRmax was 70–80% (F = 1.000, $p = 0.415$, ES = 0.408), and the %HRmax was 80–90% (F = 0.144, $p = 0.932$, ES = 0.153). These values were not significantly different.

## The effect of different loads of sled after resistance sprint on the ice push sled performance

As shown in Table 4, a one-way repeated measures ANOVA on the ice push sled performance at the four tests showed that there was a significant difference among different tests (F = 15.911, $p = 0.000$, ES = 1.628) (No BMRS: 5.08 ± 0.27; 25% BMRS: 5.05 ± 0.29; 50% BMRS: 5.02 ± 0.27; 75% BMRS: 5.04 ± 0.28). *Post hoc* comparisons showed that 50% of body mass resisted sled sprint test had significantly faster ice push sled speed than no resisted sled sprint test ($p < 0.05$), 25% of body mass resisted sled sprint test ($p < 0.05$) and 75% of body mass resisted sled sprint test ($p < 0.05$). In addition, 75% of body mass resisted sled sprint test had significantly faster ice push sled speed than no resisted sled sprint test ($p < 0.05$).

# DISCUSSION

The morning pulse, morning BU, and CK were all tested on the day of the test. Our results showed that there was no significant change in the physical function of the athletes on the

**Table 2 Morning pulse, morning BU, and CK test results before different loads resisted sled test (M ± SD).**

| Factors | NO BMRS | 25% BMRS | 50% BMRS | 75% BMRS |
|---|---|---|---|---|
| Morning pulse/(beats·min$^{-1}$) | 54.29 ± 6.02 | 54.43 ± 4.89 | 53.86 ± 6.82 | 54.14 ± 6.47 |
| BU/(mmol·L$^{-1}$) | 6.42 ± 0.42 | 6.39 ± 0.50 | 6.52 ± 0.49 | 6.47 ± 0.43 |
| CK/(U·L$^{-1}$) | 155.43 ± 43.42 | 154.86 ± 49.16 | 160.71 ± 41.38 | 156.43 ± 35.86 |

Note:
**BMRS,** Body mass resisted sled; BU, blood urea; CK, creatine kinase.

**Table 3 Heart rate monitoring results in warm-up phases of different loads resisted sled test (M ± SD).**

| %HRmax | No BMRS/min | 25% BMRS/min | 50% BMRS/min | 75% BMRS/min |
|---|---|---|---|---|
| ≤60% | 12.71 ± 3.95 | 11.86 ± 3.02 | 13.43 ± 4.20 | 12.57 ± 3.10 |
| 60–70% | 14.29 ± 3.59 | 14.43 ± 3.55 | 13.57 ± 2.44 | 13.57 ± 2.82 |
| 70–80% | 8.14 ± 2.85 | 9.43 ± 3.74 | 8.57 ± 2.94 | 9.29 ± 2.43 |
| 80–90% | 4.86 ± 4.18 | 4.29 ± 2.69 | 4.43 ± 2.37 | 4.57 ± 2.15 |
| 90–100% | 0.00 ± 0.00 | 0.00 ± 0.00 | 0.00 ± 0.00 | 0.00 ± 0.00 |

Note:
**HRmax,** Maximum heart rate; BMRS, body mass resisted sled.

**Table 4 Ice push sled performance after resisted sled sprints with different loads.**

| Participant | No BMRS/s | 25% BMRS/s | 50% BMRS/s | 75% BMRS/s |
|---|---|---|---|---|
| 1 | 4.82 | 4.79 | 4.78 | 4.77 |
| 2 | 4.74 | 4.70 | 4.69 | 4.72 |
| 3 | 4.80 | 4.76 | 4.74 | 4.76 |
| 4 | 5.30 | 5.30 | 5.25 | 5.28 |
| 5 | 5.37 | 5.38 | 5.30 | 5.35 |
| 6 | 5.25 | 5.19 | 5.18 | 5.20 |
| 7 | 5.25 | 5.25 | 5.22 | 5.23 |
| M ± SD | 5.08 ± 0.27 | 5.05 ± 0.29 | 5.02 ± 0.27[abc] | 5.04 ± 0.28[d] |

Notes:
[a] A significant difference in the 50% of body mass resisted sled condition compared to the no resisted sled condition ($p < 0.05$).
[b] A significant difference in the 50% of body mass resisted sled condition compared to the 25% of body mass resisted sled condition ($p < 0.05$).
[c] A significant difference in the 50% of body mass resisted sled condition compared to the 75% of body mass resisted sled condition ($p < 0.05$).
[d] A significant difference in the 75% of body mass resisted sled condition compared with the no resisted sled condition ($p < 0.05$).

day of the test. The heart rate of the athletes was monitored during the warm-up process and the athletes maintained the same intensity of the warm-up process during the four tests. Therefore, it is possible to exclude the interference of subjects' individual factors, and the key variable in this study was the resisted sled sprint with different loads. A systematic review concluded the effect of rest time after PAP on the performance of subsequent sprints and found that a longer rest time of 5–7 min produced the greatest PAP (*Seitz & Haff, 2016*). In addition, taking into account the actual situation of skeleton events, it takes

at least 6 min for an athlete to change to a professional racing suit after warm-up and to be ready to start the race. Therefore, the rest time in this study was set at 6 min. In the present study, PAP was induced by resisted sled sprint at 75%, 50% and 25% of body weight with a recovery time of 6 min.

In this study, we found that after the 50% of body mass resisted sled sprint PAP intervention, the athletes were on average 0.06 s faster compared to the no resisted sled sprint test, and all athletes had shorter 30 m ice push sled time compared to the no resisted sled sprint test, thus the 50% of body mass resisted sled sprint PAP intervention significantly improved the athletes' 30 m ice push sled performances. In addition, the 30 m ice push sled time was significantly shorter by an average of 0.04 s after the 75% body mass resisted sled sprint compared to the no-resistance sled sprint test. These results are somewhat similar to those found by *Zisi et al. (2022)*, who reported significant improvements in subsequent 30 m sprint speed after using an individualized sled load of 57–73% of body mass in PAP interventions, and which also identified 57–73% of body mass as the optimal sled resistance load. Although the main focus of our study was to investigate the effect of resisted sled training on ice push sled rather than sprinting performance, this type of study is informative given that ice push sled and sprinting have similar mechanical characteristics (*Needham et al., 2021*) and no study has yet analyzed the effect of this type of PAP-induced program on the ice push sled performance of skeleton athletes. A previous study by *Tian et al. (2022)* also found that the 50% of body mass resisted sled training significantly improved the 30 m sprint performance of skeleton athletes, with an average time of 0.08 s faster than in the no resisted sled test. In our study, the resistance load, resistance method, and subjects were similar to that study, but the athletes pushing the sled on the ice only on average of 0.06 s faster than the no resistance test. Nevertheless, the improvement in ice push sled performance observed in this study is significant given that skeleton races have been decided in as little as 0.01 s in some cases (*Colyer et al., 2017*; *Gong, 2023*). There is even a historical precedent of two teams sharing medals (with a time difference of less than 0.01 s) (*Liang et al., 2021*). Ice push sled performance continues to positively influence skeleton race performance. Athletes who underwent PAP did not improve their ice push sled performance as much as their sprinting performance compared to the control group, which may be due to several factors. During the push sled phase of skeleton activation, athletes are typically required to quickly push a 30–40 kg sled. This resistance sprint differs significantly from a sprint in that the athlete is required to exert greater force. The initial section of the skeletal push track must have a 2% incline, after which the incline becomes steeper, and the subsequent 60 m section must have a 12% incline (*Colyer et al., 2018b*). The biological limit of sprinting speed is determined by the ability to generate the required force in a very short contact time, not just the maximum force generated by the lower limbs. Maximum running speed is higher and ground contact time is shorter on slopes compared to horizontal surfaces, so rapid force generation may be a stronger determinant of ice push sled performance in skeleton. The study by *Colyer et al. (2018b)* also noted that it may be important to increase the velocity of maximal muscle contraction to improve ice push sled performance.

Changes in ice push sled kinematics resulting from 50% and 75% of body mass resisted sled sprint may result in beneficial activation effects, with resisted sled sprint increasing the stimulation of muscular strength, peak power, and rate of power development (*Fradkin, Zazryn & Smoliga, 2010*; *Martínez-Valencia et al., 2015*). A study by *Winwood et al. (2016)* concluded that heavier loads induced greater involvement of larger muscle fibers in the specific motor units required for sprinting, which in turn induced greater neural and muscular mechanisms, resulting in an acute increase in explosive sprinting ability. In addition, resisted sled sprint with heavier loads can enable athletes to generate greater horizontal or synthetic ground reaction impulses to improve sprint acceleration (*Kawamori et al., 2014*). Some studies have found resisted sled sprint to be a potential method of increasing power output and efficiency of power output relative to unresisted sprinting (*Fradkin, Zazryn & Smoliga, 2010*; *Okkonen & Häkkinen, 2013*). The sled resistance loads used in this study were better stimulated to produce the PAP effect to improve ice push sled performance, with the following possible mechanisms: first, resisted sled sprint results in the activation of sprint-specific type II muscle fibers, leading to higher force and power production; and second, myocardia regulates light chain phosphorylation, leading to an influx of calcium to create more binding sites (*Wong et al., 2017*). Skeleton programs have been found to be fast and have high explosive power requirements (*Li, Chen & He, 2018*), and sprint acceleration and lower body strength are known to be the most important factors influencing overall sled performance in skeleton athletes (*Colyer et al., 2018b*). In terms of technical efficiency of physical performance, a resisted sled sprint increases horizontal propulsion compared to an unresisted sprint (*Okkonen & Häkkinen, 2013*). This finding may be due to the increased angle of trunk tilt during the resisted sled sprint compared to the unresisted sprint (*Cronin et al., 2008*), which allows for greater horizontal force to be applied. While skeleton athletes must demonstrate the ability to generate force at high speeds when jumping onto the sled, the ability to apply greater force horizontally may help to increase sled efficiency and thus improve on-ice push-skid performance. *Alcaraz et al. (2008)* found that resisted sled training is an effective training method for developing sprinting performance, especially during the early acceleration phase of less than 10 m. In the skeleton program, the timing point of the starting phase of the push sled is 15–65 m, and the loading point is generally 30 m (*Jing & Yuan, 2020*). Therefore, if the speed can be maximized in the 0–15 m phase of the start and maintained until about 30 m on the sled, there will be a greater advantage in the start.

In the present study, there was no significant effect on subsequent ice push sled performance after a 25% of body mass resisted sled sprint, which is consistent with the team's previous research. There was no significant effect on subsequent 30 m sprint performance after a 25% of body mass resisted sled sprint (*Tian et al., 2022*), but few studies have examined the effect of PAP on skeleton ice push sled performance. *Smith et al. (2014)* found that lighter sled loads may be less suitable for improving acute sprint performance. Their study used a 25–30% body mass sled for three resisted sprints, which not significantly improve sprint speed, leading them to conclude that sled loads may not be sufficient to induce PAP. *Whelan, O'Regan & Harrison (2014)* found that resistance sprinting with sled loads of 25–30% of body mass did not demonstrate PAP for 10 m sprint

performance in the presence of resistance. In conclusion, the reason that 25% of body mass resisted sled sprints are not sufficient to elicit PAP may be that there is not enough stimulus and the PAP response is unlikely to occur.

## Strengths and limitations

### Strengths

The present study belongs to the first PAP intervention on the starting ice push sled performance in the sport of skeleton. (1) Experimental subjects and practical applicability: The study selected key athletes of the Chinese national team in skeleton as the research subjects, who are the core members of the team preparing for the Winter Olympic Games and have high level and practical experience. Therefore, the results of the study have strong practical applicability and can provide guidance for the training and competition strategies of the Chinese skeleton team. (2) Exploratory application of the PAP strategy: The study used resisted sled sprint exercises with different loads to induce the PAP effect, and evaluated the impact of PAP by testing the changes in the performance of pushing sled on ice. This exploratory application can provide new ideas and guidance for the adoption of PAP strategy in training and help athletes to improve their on-ice sled pushing performance.

### Limitations

The seven key athletes of the Chinese skeleton team were selected for this study, and although these players possessed a high level of performance, the sample size was small. Therefore, the generalizability and replication of the results are somewhat limited, limiting the extrapolation of the results to the general population. Future studies could increase the sample size and expand the scope of the study to better validate and generalize the results. The non-randomized test sequence of this study may lead to potential effects of learning effects and psychological adaptation. As the tests progressed, subjects may have improved their performance by increasing their skill or proficiency, which was particularly evident in the lower loaded tests. In addition, subjects' psychological states and expectations following higher loaded tests may influence their performance on lower loaded tests. These factors may lead to biased results and affect the accuracy and comparability of the data. However, we required at least 48 h between each test in our study, which minimized the interference of learning effects and psychological adaptation on the data. Although the same 20 m distance was used in all experimental tests, it should be noted that the times under tension were not fully standardized. In particular, the 20 m unloaded time was significantly shorter than the time under the heavier load test. Therefore, if there is a difference in overall stretching times across tests, this may be due to the different sprinting effects of the same sprint time over the same distance. This factor should be considered a limitation of the study.

## CONCLUSION

A single 20 m sprint with a 50% of body weight resisted sled is effective in increasing the PAP effect in skeleton athletes, which in turn improves their ice push sled performance.

Therefore, coaches and athletes can use a 50% of the athlete's body weight resisted sled in an all-out sprint prior to daily training and competition to improve the ice sled push performance of skeleton athletes in competition.

### Funding

This study was supported by the Research and Applications of Key Techniques for Eliminating Sports Fatigue in Olympic Winter Sports (2019YFF0301603) and the Shanghai Key Lab of Human Performance (Shanghai University of sport) (No. 11DZ2261100). The funders had no role in study design, data collection and analysis, decision to publish, or preparation of the manuscript.

### Grant Disclosures

The following grant information was disclosed by the authors:
Research and Applications of Key Techniques for Eliminating Sports Fatigue in Olympic Winter Sports: 2019YFF0301603.
Shanghai Key Lab of Human Performance: 11DZ2261100.

### Competing Interests

The authors declare that they have no competing interests.

### Author Contributions

- Guang Tian conceived and designed the experiments, analyzed the data, prepared figures and/or tables, and approved the final draft.
- Haojie Li conceived and designed the experiments, authored or reviewed drafts of the article, and approved the final draft.
- Huan Zhu performed the experiments, analyzed the data, prepared figures and/or tables, and approved the final draft.
- Binghong Gao performed the experiments, authored or reviewed drafts of the article, and approved the final draft.

### Human Ethics

The following information was supplied relating to ethical approvals (i.e., approving body and any reference numbers):
The Ethics Committee of Shanghai University of Sport (No. 102772020RT081).

### Data Availability

The raw data is available in the Supplemental File and figshare: Tian, Guang (2024). Raw statistics. figshare. Dataset. https://doi.org/10.6084/m9.figshare.25524460.v2.

### Supplemental Information

Supplemental information for this article can be found online at http://dx.doi.org/10.7717/peerj.18271#supplemental-information.

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
