# Peer review of "The effect of post-activation enhancement on the performance of Chinese national skeleton athletes in the “ice push sled”-a first cross-sectional study"

_PeerJ, doi:10.7717/peerj.18271_

## Round 0.1 · original submission · Major Revisions

Dear Dr. Tian,

Your manuscript titled "The Effect of Post-Activation Enhancement on the Performance of Chinese National Skeleton Athletes in the "Ice Sled Push" - A First Cross-Sectional Study" was considered by two expert reviewers and based on their opinions and my review, the decision is “Major revision”.

Please carefully read the reviewers’ comments and address them fully in your revised manuscript. In addition, please address the following points:

General comments:

(1) Please review the entire manuscript as there are multiple errors of truncated sentences, repeating sentences and other syntax issues.
(2) Seeing that the sample number is small (n=7) is an ANOVA test valid? Did the authors test their samples for normal distribution before running an ANOVA test?
(3) Results are presented in the abstract differently than in the main text (compare L22-4 to L166-70).
(4) Looking at table 4, the difference between groups is miniscule (e.g., between 25% and 50% BMSR it is 30 milliseconds). Together with the very small sample size (n =7), I wonder if the statistically significance difference has any real biological meaning (see L181-3); despite what the authors wrote in L188-93. The authors must, at least, address that as a limitation.
(5) Please explain what “ES” values are (ANOVA test, L150, L157-9 and more).
(6) The reference list is confusing with multiple styles, missing data and repeats (e.g. references 6 and 8, references 16 and 26 and more).
(7) References in text are not in the right format.

Specific comments:
- L123-4: "... and an average ice temperature of -8.1°C. The average ice temperature was -8.1°C and the average ice temperature was -8.1°C.". The same info is repeated 3 times.
- L151: this line/sentence is a repeat and should be deleted.
- L169: “…75% body mass sled resistance group…”. This sentence is a repeat.
- L178: “The key variable in this study was whether to include the sled resistance sprints with different loads”. I don’t think this sentence describes what the authors have intended. Please rewrite.
- L183-5: I’m not sure if the authors cite reference #30 correctly. It seems that in that study all groups were 50% body mass (what different was the number of repetitions 1, 2 or 3).
- L242: “This study belongs to the first PAP intervention analysis of skeleton sport”. I don’t think this sentence describes what the authors have intended. Please rewrite.
- Figure 1 is not clear as it presents progression horizontally and vertically, but it is not clear how these 2 axes correlate.

Please ensure that all review, editorial, and staff comments are addressed in a response letter and any edits or clarifications mentioned in the letter are also inserted into the revised manuscript where appropriate.

Please note that submitting a revision of your manuscript does not guarantee eventual acceptance, and that your revision may be subject to re-review by the reviewer(s) before a decision is rendered.

·

Basic reporting

The manuscript is clearly written in professional English language.
Figure and Tables are relevant and described. Moreover, the Raw data shared.
Intro and background of skeleton event are supported. However, some references didn't met with the sentences which referred through the manuscript.
Furthermore, the introduction need studies to further support the strength of the current study (explaining why differ/better than previous studies on the field).
The weakness of this study is that confusing with some mix statements based on land running sprint and others based on ice-sled sprinting studies. These statements should be clarified based on each type of studies to avoid misinterpretations.
The references didn't met the publication criteria, it should be corrected.

Experimental design

This study is original, and the main strength is to address an interesting practical question in high level athletes about skeleton performance in daily training and during competitions. The method have been used for the study design are valid but the examined variable (ice push sled test) is not clearly described. It should be clarified more specifically the testing procedure.
Additionally, there is a limitation which is important to be referred about the overall time under tension among conditions.

Validity of the findings

The discussion section partly support the study findings because of the confusing statements based on running sprint which mixed with ice-sled sprinting studies.
Recommend to present more the uniqueness and benefits of the study. In the conclusion included the practical application and the potential benefits of the current study for the training coaches and practitioners.

Additional comments

Thank you for your submission

1. Line 13 and throughout the text: the word "optimal" is misleading and has been wrongly interpreted sometimes. It suggests that this individual load is the best for all types of mechanics and performance improvements where in fact it is only associated with the apex of the velocity-power relationship in sprinting. I would remove "optimal" because in the current study the loads is not associated with the apex of the individual velocity-power relationship but according to individual body mass. This would avoid misinterpretations of the word "optimal" by uninformed readers.

2. Line 17 and throughout the text: “25% of body mass” instead of “25% body mass”. Use “of” before body mass when you referred about the loads.

3. Line 22 and throughout the text: The sentence “group” should be changed to “conditions” because the same participants complete all the testing conditions (no resistance and with loads of 25, 50 and 75% of Body Mass).

4. Line 22-24: In the results, it is more important to see mean differences ± and SD changes observed, so please update.

5. Line 23: It is referred that “the 50% body mass sled resistance group had lower performance compared to the no resistance”. On the other hand at line 27 it is referred that with the current load improved sprinting performance. "Lower performance" means that the performance decreased with the load of 50% of body mass, so please use “greater” instead of “lower” or use “increased/improved” (or something similar) and rephrase the text.

6. Line 32: It is referred that “It is characterized by high speed and high risk” (Davenport et al. The Appropriateness of Snowmobiling in National Parks: An Investigation of the Meanings of Snowmobiling Experiences in Yellowstone National Park. Environmental Management, 2005) I’m not pretty sure that this reference is met with your sentence. Please check and correct.

7. Line 40: The reference with number 8 (Min et al. 2022) is the same reference with number 6. You should also correct the author name at the references section.

8. Line 45-46: The reference with number 12 (QinLong et al. 2022) referred to the text as follows: “the success of the U.S. skeleton team in recent Winter Olympics was partly due to the athletes' excellent sled technique, which enabled them to maintain a leading position on the track”. ) I’m not sure that this reference is met with your sentence. Please check that this reference is correct.

9. Line 63: In the entire introduction, make sure not to mix statements based on land running sprint and others based on ice-sled sprinting studies. Please clarify the statements based on each type of studies.

10. Line 75: “kg” instead of “kilograms”. It is recommended to use the SI standard.

11. Lines 78: It is referred that “there are very few studies on the effect of PAP on the starting push performance after sled resistance sprinting”. However these studies isn’t reported. The paragraph need studies to further support the strength of the current study (explaining why differ/better than previous studies on the field). Please update.

12. Lines 109-112: In all conditions used the same 20-m distance. However, the time under tension is not perfectly standardized as a 20-m is much shorter (in seconds) with no load compared with heavier loads. Please clearly acknowledge this as a limitation if the overall time under tension (if the same sprint time sprints over the same sprint distance) wasn’t equal between conditions.

13. Line 112: Figure 1: It is suggested to be written as follows: First condition - control or no resistance, Second condition - 75% of BM sled resistance.. etc. instead of The first time, second time, third time and fourth time. Moreover, “Ice push sled test” instead of “Ice push Pry test”.

14. Lines 117-124: “The ice push sled test” isn’t clearly defined. It should be clarified more specifically the testing procedure.

15. Line 123-124: The sentence “average ice temperature of -8.1°C” repeated for 3 times, please correct.

16. Line 151: “0.316). These values were not significantly different.” Suggested to be deleted because it was referred above.

17. Line 158: Add space between “was” and “70-80%”.

18. Lines 164-170: “condition(s)” instead of “group(s)”.


19. Lines 168-170: The sentence: “The 25% body mass sled resistance group (P<0.05), 75% body mass sled resistance group (P<0.05), and the 75% body mass sled resistance group had significantly lower ice sled push performance than the no sled resistance group (P< 0.05).” is not clear to me. First, the sentence “the 75% body mass sled resistance group” repeated twice. Second, as you referred in the table 4 the 50% body mass sled resistance group significantly increased performance, compared to 25 and 75% conditions and the load of 75% of BM significantly improved performance compared to control condition. Please clarify the results and update the text.

20. Line 167 and 169: “greater” instead of “lower”.

21. Line 170: (P<0.05) instead of (P< 0.05)

22. Line 172 Table 4: Please use the alphabetic symbols a, b, c.. to highlight the significant differences instead of #, @, &.

23. Line 185-188: In these lines referred that “The results of this study are similar to those of Paul [30], whose researchers reported significant time improvements in the subsequent 15-meter sprint after using a sled loaded with a 50% body mass PAP protocol”. At first, the reference is not “Paul” but Jarvis et al. (Jarvis Paul). Second, this research conducted on land surface and not on ice so the sentence “similar” is not correct at these lines. To avoid misinterpretations make sure not to mix statements based on running sprint and others based on ice-sled sprinting studies. Please clarify the statements based on each type of studies.

24. Line 185-186: According to Zisi et al. the optimal loading associated with the individual load that causes a velocity decrement of 50% and referred to the individual’s apex of the power-velocity relationship during sprinting on a land surface (see also comment 1). Additionally, Cross et al, Stavridis et al. and Lahti et al. (Optimal loading for maximizing power during sled-resisted sprinting. Int J Sports Physiol Perform 2017; The Effects of Heavy Resisted Sled Pulling on Sprint Mechanics and Spatiotemporal Parameters. JSCR 2023; Changes in sprint performance and sagittal plane kinematics after heavy resisted sprint training in professional soccer players. PeerJ 2020, respectively) reported a range of loads which causes a velocity decrement of 50%: from 69 to 91% of BM for recreational sprinters; from 58 to 72% of BM in well trained sprinters and from 70 to 96% of BM to high level soccer players. According to these references, it should be clearly defined that all these studies conducted on land and not on ice and the current load describes an individual load according to the force-velocity relationship, the different level of practice, sex and different sports. Additionally, these studies conducted with a pulling and not pushing sled. There is differences in loads (in kg of body mass) when you pull or push of a sled in order to perform the same velocity decrement (please see Cahill et al. Resisted Sled Training for Young Athletes: When to Push and Pull. Strength and Conditioning Journal 2020).

25. Line 207: Add a space between reference number “35” and “A study”.

26. Lines 211-213: The sentence “Some studies have found sled resistance sprinting to be a potential method of increasing power output and efficiency of power output relative to unresisted sprinting” is not supported with references, please refer these studies that you mentioned.

27. Lines 219-221: In these lines referred that “a skeleton resisted sprint increases horizontal propulsion compared to an unresisted sprint.” However, according to the reference number 40 (Biomechanical comparison between sprint start, sled pulling, and selected squat-type exercises) Okkonen et al. only examined the kinetics and kinematics of a sprint start from blocks and not during skeleton event. Please correct this reference or remove the word “skeleton” from the sentence.

28. Lines 236-237: In these lines referred that “Lowery et al [45] showed that moderate and high intensities resulted in PAP, whereas lower intensities did not.” However, Lowery et al. examined only the vertical jump performance and not the sprinting performance as you mentioned about the different intensities. Please clarify that these intensities are not for sprinting performance.

29. Lines 265-377: References it should be corrected, in some cases authors names is first and in other cases the authors surnames. It is recommended to use an automatic citation program.

Reviewer 2 ·

Basic reporting

This study investigated the effect of PAP on snowmobile sprint starts. This is an interesting topic as described by the introduction, and overall, the study has merit but there are some areas outlined below where need further explanation is required to make it more rigorous.


1. Although the study has the strength of examining seven elite athletes, the weakness is that the study may be underpowered. To overcome this having the statistical power noted in the methods section would be welcomed.

2. Can you provide more detail of the ice sled tests – were they performed on an ice course. You have stated they were performed on a plastic track igloo; I assume this is ice? Also, you need to detail what equipment the participants were wearing, I assume it was their racing equipment and can you indicate clothing and shoes etc.

3. Table 1 - Genders should be spelt gender. There is no explanation of what IC is can you provide an explanation at the end of the table.


4. Table 2 – Insert the note of the abbreviations at the end of the table.

Experimental design

5. Line 104-105 - The warm-up has been described, but to further enhance this it would be appropriate to detail the of where the jogging occurred, the muscle groups targeted for the dynamic stretching, the dynamic sprint exercises that were prescribed along with the number of sets, duration and intensity.

6. Line 107 – It is not clear how the sled mass was increased to 25%, 50% and 75% of body mass. How was this external load placed on the sled. Can you provide a full description of how this was done.

7. Line 108 – It is not clear if the sequence of testing was randomised for the various resistance loads. If it was not randomised, it is a major weakness of the study but can be overcome by stating this in the methods section that the order of testing was performed from 75% first, followed by 50% and finishing with 25%. Additionally, you need to acknowledge this as limitation and provide a narrative of how it could have influenced the results and this should be reported in the discussion section.

8. Line 108 – Another weakness of the study is the rationalisation in selecting a 6 minute rest. The PAP response may have occurred before or after 6 minutes. Previous research suggests that PAP can be effectively induced through a recovery period ranging from 3 to 9 minutes. For your study you need to validate why you selected 6 minutes and state this explicitly with the appropriate literature.

9. Line 117 – what model of Omega timing device was used? It is part of the course? And describe the start and end points along the track where the timing was performed.

10. Line 121 – you have written the average ice temperature twice – please delete.

11. Line 124 – It is not clear on the rationale of measuring blood urea, creatine kinase for this study. Can you provide a detail justification of why these measures were used for the study and what’s the relevance to PAP.

12. Line 129 – Likewise, with HR what is its purpose? Can you provide and explanation of why you have included this measure for your study.

13. Line 135 – please include SPSS location of city and country of origin.

14. Line 139 – spell out LSD

15. Line 140 – lower case ‘p’

16. Line 153 – It is not clear how % HR max was determined. I suggest in the methods sections that you clearly define this.

17. From the results of BU, CK and HR it is not clear on why they were collected and how it relates to warm-up and PAP. As previously suggested it is important to rationalise these variables in the methods.

18. Line 162 – lowercase ‘p”

Validity of the findings

19. Line 163 – you state that “the 50% body mass sled resistance group had significantly lower ice sled push sprint performance than the no sled resistance group (P<0.05).” Firstly, lower case for the p value, secondly it would be advantageous to state the time difference of the this significance. Likewise for line 165, including times would be advantageous. To remove ambiguity, it would be better to use wording of faster or slower, rather than lower or higher. This also should be reflected in the abstract. Ensure all p values are in lower case.

20. Table 4 – Ensure p values are lower case. It is not clear what the difference columns are for, therefore a label is required, or I would suggest removing all of them. This will allow the reader navigate the table a lot easier. If time differences need to be included across the different resistance loads, I suggest creating a new table for this. Also, to improve readability, I suggest that the symbols you use for significance are in superscript.

21. Line 180 – remove ‘ whose researchers’ replace with who

22. Line 185 – delete ‘and subjects remained the same,’

23. Line 246 – It’s good that you have identified the small sample size, but it would be good to report the actual statistical power and include if the study was randomised and the selection of 6 minutes rest could have missed the PAP response. Also included split times at 5 m, and 10 m would also provide valuable insight to the acceleration phase of the 20 m sprint.

Additional comments

24. Abstract – you state ‘ Post hoc analyses revealed that the 50% body mass sled resistance group had lower performance compared to the no resistance, 25% resistance, and 75% resistance groups. Additionally, the 75% resistance group had lower performance than the no resistance group’. To strengthen this further you could add the time differences that were noted as significant. This will provide some context to the results.

---

## Round 0.2 · Minor Revisions

Dear Dr. Tian,

Your revised manuscript, titled "The Effect of Post-Activation Enhancement on the Performance of Chinese National Skeleton Athletes in the "Ice Sled Push" - A First Cross-Sectional Study", was evaluated by two expert reviewers and based on their opinions and my review, the decision is “Minor revision”.

Please carefully read the reviewers’ comments and address them fully in your revised manuscript. In addition, please scrutinize your manuscript for typos, repetitions, inconsistencies and syntax issues (Note: line numbers refer to tracked changes document):
(i) Typos (e.g., spaces are missing between the text and the opening parenthesis of in-text citations, L193 has an extra period, L368 “programme”, L388 (and other locations) “resisited”, L425 double spacing, etc.)
(ii) Repetitions (e.g., L207-11 repeats the same info twice)
(iii) Inconsistencies (e.g., using “meters” vs. “m”, “minutes vs. “min”, using “test” and “condition” to describe the same thing, using “s” after a number with, or without a space, L18 “48-hours” vs. L168 “48 hours”, etc.)
(iv) Syntax (e.g., L360)

Please ensure that all review, editorial, and staff comments are addressed in a response letter and any edits or clarifications mentioned in the letter are also inserted into the revised manuscript where appropriate.

Please note that submitting a revision of your manuscript does not guarantee eventual acceptance, and that your revision may be subject to re-review by the reviewer(s) before a decision is rendered.

·

Basic reporting

I would like to thank the authors for addressing all my suggestions.
The revised manuscript has been totally improved.
I overall think this paper is worth being published as it addresses a novelty topic about PAP intervention on the starting ice push sled performance in the sport of skeleton.
I only have some minor comments to suggest:

Line 141: duplicate text
Line 173: duplicate text
Line 219: there is a symbol instead of the Reference.

Experimental design

no comments

Validity of the findings

no comments

Additional comments

no comments

Reviewer 2 ·

Basic reporting

A final check on grammar and syntax is required. Additionally, ensure SI units are consistently used for example line 18 it reads "20-meter sprints" it should be 20 m.

Experimental design

My responses have been adequately addressed.

Validity of the findings

My responses have been adequately addressed.

Additional comments

My responses have been adequately addressed.

---

## Round 0.3 · accepted · Accept

Dear Dr. Tian,

After careful consideration of your revisions and the reviewers’ comments, we have determined that your work meets our publication standards. Thus, I am pleased to inform you that your revised manuscript, "The Effect of Post-Activation Enhancement on the Performance of Chinese National Skeleton Athletes in the "Ice Sled Push" - A First Cross-Sectional Study", has been accepted for publication.